# Local and Traditional Ecological Knowledge of Fish Poisoning in Fiji

**DOI:** 10.3390/toxins15030223

**Published:** 2023-03-16

**Authors:** Jimaima Veisikiaki Lako, Sereima Naisilisili, Veikila C. Vuki, Nanise Kuridrani, Dominic Agyei

**Affiliations:** 1School of Applied Sciences, College of Engineering, Science and Technology, Fiji National University, Suva P.O. Box 15676, Fiji; 2School of Education, Faculty of Arts, Law and Education, The University of the South Pacific, Laucala Campus, Private Mail Bag, Suva, Fiji; 3Oceania Environment Consultants, UOG Station, Mangilao 96923, Guam; 4Ministry of Fisheries, Lami Station, Suva P.O. Box 3165, Fiji; 5Department of Food Science, University of Otago, Dunedin 9054, New Zealand

**Keywords:** fish poisoning, ciguatera, traditional ecological knowledge, toxic fish species, marine toxin hotspots

## Abstract

Fish poisoning (FP) affects human health, trade and livelihood in Fiji, where management has depended mainly on traditional ecological knowledge (TEK). This paper investigated and documented this TEK through a 2-day stakeholder workshop, group consultation, in-depth interviews, field observations, and analyses of survey data from the Ministry of Fisheries, Fiji. Six TEK topics were identified and classified as preventative and treatment options. The preventive approach involves identifying toxic reef fishes, the spawning season of edible seaworms, hotspot areas of toxic fishes, folk tests, and locating and removing toxic organs. For example, 34 reef fish species were identified as toxic. The FP season was associated with the spawning of *balolo* (edible seaworm) and the warmer months of October to April (cyclone seasons). Two well-known toxic hotspots associated with an abundance of *bulewa* (soft coral) were identified. Folk tests and locating and removing toxic fish organs are also practised for moray eels and pufferfish. At the same time, various locally available herbal plants are used to treat FP as the second line of defence. The TEK collated in this work can help local authorities better identify the sources of toxicity, and applying TEK preventive measures could stem the tide of fish poisoning in Fiji.

## 1. Introduction

Local and traditional ecological knowledge (TEK) is recognised as a legitimate knowledge system and used in various areas such as the conservation, management, and restoration of degraded environments and ecosystems [1], management of artisanal fisheries [2], identification of shark river habitats [3], and in the management and treatment of ciguatera poisoning (CP) [4,5,6,7].

Local knowledge or folk science held by individuals or groups is based on personal or cultural experiences and observations. These systems of knowing describe ways of understanding and predicting the natural and social environment but are not necessarily based on rigorous ‘scientific’ or empirical observation [8]. TEK is, therefore, the cumulative body of knowledge, practice, and beliefs that evolve through adaptive processes and are handed down through generations by cultural transmission [1]. TEK usually revolves around indigenous natural resources in a community and is created to safeguard these resources. For example, there are several TEK around marine resources such as fish in the Pacific Island Small States [9]. However, in the Pacific Island Small States, there is a growing incidence of fish poisoning (FP), which is negatively impacting fish quality/safety, human health, and economic security.

Fish, an important protein-rich food source that contributes significantly to the health and economic livelihood of most Pacific Islanders, has an annual consumption of between 13 and 145 kg per capita [10,11]. In Fiji alone, fish consumption is estimated at between 15 and 113 kg per capita, and there is a higher per capita consumption in the rural (25 kg) and coastal communities (113 kg) compared to the urban centres (15 kg) [10,12]. A study by Kuster, Vuki, and Zann (Ref. [13]) on fish consumption in Ono-i-Lau Island in the Lau Islands group of Fiji revealed that fish remained the major source of protein between 1982 and 2002. The fact that fish plays an important role in the food system and economy of Pacific Islanders means that the presence of toxins in fish will have a huge impact on the economy and public health of the Islanders, especially when it involves fish species destined for export or the hospitality industry.

The incidence of FP in the Pacific region has been increasing [14]. For example, in Fiji, a 207% increase over 10 years between 2008 and 2017, with a total incidence of 15,114 cases, was reported by Holbrook et al. [15]. Despite this, fatalities in Fiji are uncommon, except in 2017, where 4 died out of 13 people who were poisoned after eating bluestripe herring (*Herklotsichthys quadrimaculatus*) (https://fijisun.com.fj/2017/01/06/4-dead-9-under-observation/ (accessed on 17 December 2022)). A report of a young couple who might have died from CP in Fiji is reported in Mermel [16].

Tetrodotoxins, palytoxin, and ciguatoxin are among the most popular toxins responsible for FP in humans. These toxins are heat resistant and so cannot be degraded by thermal treatments such as boiling, baking, frying, or freezing [17,18].

Pufferfish poisoning is caused by the consumption of tetrodotoxins, a potent neurotoxin found in fish from the order Tetraodontiformes [19]. Palytoxin is a lethal toxin, mainly present in zoanthid corals; certain marine organisms, including fish, feed on these palytoxin producers, as well as on benthic dinoflagellates, especially the genus *Ostreopsis* [20,21].

Ciguatera poisoning (CP) arises from consuming seafood contaminated with ciguatoxins and is the most common type of seafood poisoning in Fiji [22]. Incidentally, the Food Agricultural Organisation (FAO) is currently setting up a ciguatera surveillance and monitoring programme for Fiji, Tonga and Samoa.

Tropical reef fish species such as barracudas (*Sphyrnidae sphyrna*), mangrove red snappers or mangrove jacks (*Lutjanus argentimaculatus*), moray eels (*Gymnothorax* spp.), emperors (*Lethrinus* spp.), and groupers (*Epinephelus* spp.) are common fish species implicated in the incidences of CP [23,24]. There are about 175 different symptoms that have been recorded in both the acute and chronic phases, with major symptoms including gastrointestinal (diarrhea, nausea, vomiting, abdominal pain), cardiovascular (bradycardia, hypotension) and neurological disturbances (circumoral and general paresthesia, dysesthesia, such as cold allodynia, itching, muscle weakness, and asthenia), appearing within 48 h after the ingestion of a toxic meal [18,23].

Although FP is prevalent in the Pacific region, not all countries require mandatory reporting of CP. For example, in Fiji, FP surveillance and reporting are not mandatory, and the epidemiological FP data are recorded by the Ministry of Health and Medical Services. These data are gathered from FP incidences that visited the hospital but do not distinguish the types of FP. Studies show that there are more cases of FP in rural and isolated maritime communities which are not reported due to isolation and lack of resources [5,14]. Therefore, the true incidence of ciguatera is difficult to ascertain due to significant under-reporting of the illness, failure to recognise the symptoms, and limited epidemiological data collection. These factors are why it is believed that only 2–10% of CP cases are reported to health authorities [17,22].

To date, there is no reliable treatment or antidote for FP, including CP, tetrodotoxin and palytoxin poisoning. Therefore, chronic FP cases provide most of the data for epidemiological assessments. Intravenous mannitol is often recommended to treat CP, but its efficacy has been disputed [18].

In Fiji and other Pacific Island countries, the management of FP over the years has mainly depended on local knowledge or folk science and TEK. These TEK need to be formally documented because they are useful to fishers, communities, the FP victims, the government, and researchers in the effort to prevent, manage, treat, and reduce the incidence of FP, especially now with the impact of climate and ecological changes that may have contributed to the widespread of ciguatoxic benthic dinoflagellates. The use of traditional herbal remedies to treat FP has been reviewed and documented in some countries [4,6] but not in Fiji. This study, therefore, investigates and documents the use of local and TEK in the prevention, management, and treatment of FP in Fiji.

## 2. Results

Fish poisoning is referred to as “*gaga ni ika*” in Fijian, which translates to “toxins in fish” or “*ika gaga*”, which translates as “toxic fish”. In Fiji’s traditional context, there is no distinction between the various FP types (i.e., whether from histamine, tetrodotoxin, palytoxin, or ciguatoxin). The type of FP is often identified by the fish species consumed. 

The results of this study revealed six local and TEK that have been used in the management of FP. These include (1) prior knowledge of perceived toxic fish species, (2) seasonality of perceived toxic fishes, (3) potential hotspots of perceived toxic fish, (4) folk tests that detect perceived toxic fish, (5) locating and removing perceived toxic fish organs, and (6) using herbal medicines to treat fish poisoning. 

### 2.1. Potential Toxic Fish Species

A total of 34 potential toxic fish species were identified and implicated in the FP incidence (see Table 1).

The commonly perceived toxic fish species were Moray eels (*G*, *javanicus* and *G. flavimarginatus*), two-spot red snapper (*L. bohar*), smoothtooth/long-face emperor (*L. Olivaceus* and *L. microdon*), blubberlip/rivulated snapper (*L. rivulatus*), white-spotted grouper (*E. caevuleopunctatus*), pick handle barracuda (*S. jello*), Russell’s snapper (*L. russelli*), leopard coral grouper (*P. leopardus*), gold spot/bluestripe herring (*H. quadrimaculatus*), camouflage/white-spotted grouper (*E. polyphekadion* and *E. coeruleopunctatus*), and mangrove red snapper (*L. argentimaculatus*). These 14 potentially toxic fish species appear to belong to six fish families: Muraenidae, Lutjanidae, Lethrinidae, Serranidae, Sphyraenidae, and Clupeidae, respectively. Moreover, a large proportion (79%) of the commonly perceived toxic fish species were found to be carnivorous fish (see Table 2).

Table 3 shows the fishes connected with the incidence of poisoning from the 2010–2015 Ciguatera Survey Data. It is worth mentioning that this data were taken from 43 districts (about 140 villages), which only covers a third of the 410 *iqoliqoli*. This Ministry of Fisheries (MoF) data shows that the most reported toxic species is *L. argentimaculatus*, which was reported in 32 out of the 43 districts. Moray eels (*G. javanicus* and *G. flavimarginatus*) were the next most reported fish in the surveyed districts.

### 2.2. Seasonality of Potential Toxic Fish

As shown in Table 1, the study responders reported that some fish species become toxic in certain seasons, while others are poisonous all year round. Moreover, it was highlighted at the stakeholders’ workshop and confirmed by interviewees that the seasonality of toxic fish species appears to be more common for CP, which coincides with the spawning of *balolo (Palolo viridis*—an edible polychaete sea-worm). *Balolo* is a delicacy, cooked in various ways and consumed in coastal communities. The spawning season for *balolo* is usually from October or November annually. In the Fijian calendar, October is regarded as the “*vulai balolo lailai*”, which is the first appearance of edible sea-worms, and November is regarded as “*vulai balolo levu*”, which is the larger appearance of edible sea-worms. *Vulai balolo lailai* usually signals the beginning of the warmer months and the cyclone season that begins in November and ends in April. 

The first appearance and harvest of *balolo* warn people to reduce and stop the harvest and consumption of known potentially toxic fish and to refrain from fishing in certain known potentially toxic hotspots. However, some fishers disregard this warning and continue to harvest, sell, and consume known toxic fish species, especially moray eels. Fishing during the perceived ‘toxic season’ demonstrates a high risk-taking tendency of fishers—a behaviour which is probably encouraged by fishers’ confidence in their knowledge of and use of TEK.

### 2.3. Iqoliqoli with Perceived Ciguatoxic Hotspots

There are 410 traditional fishing grounds (*iqoliqoli*) in Fiji. These cover approximately 30,012.41 km^2^ within Fiji’s exclusive economic zone. The *iqoliqoli* are categorised based on the district that owns the adjacent dry land, which runs parallel to the marine area. While dry lands are divided and owned by traditional sub-clans or *mataqali*, *iqoliqoli* is communally owned by the *yavusa* (traditional clans), districts or villages and is freely accessed by everyone who lives in the community. 

Stakeholders and fishers claim that there are ciguatoxin-localised reefs or ciguatoxin *iqoliqoli* hotspots where most fishes harvested are perceived toxic. Two major reefs, *Senimuna* located in *Tavuki* in Kadavu, and *Kabara* in the Lau group (Eastern Fiji), were identified as highly perceived ciguatoxin hotspots. The *Senimuna* reef, shown in Figure 1, is also confirmed by the Ministry of Fisheries (MoF) as a perceived ciguatera hotspot site, and all fish species in this reef are potentially toxic all year round. 

Further, stakeholders and interviewees reported that *bulewa* is associated with FP. They claimed that “fish species that feed on *bulewa* and are caught in areas where *bulewa* is abundant are often toxic”. For instance, *G. flavimarginatus* are usually found and caught where *bulewa* is abundant. A field visit by the Research team to the *Senimuna* reef discovered an abundance of various colourful *bulewa* on the reef with patches of bleached corals. However, no scientific investigation was carried out to confirm the presence of any toxins. 

### 2.4. Folk Tests That Detect Suspected Toxic Fish

Table 4 shows the various local practices used by coastal communities to avoid the risk of eating suspected toxic fish. These include cooking fish with silver coins and observing or assessing the discolouration of coins or any part of the fish; examining food avoidance by flies; and feeding dogs and cats with suspected fish and observing sickness or fatality in these animals. All these native, local, folk and traditional test methods of detecting toxins in suspected toxic fish species were gathered both from the stakeholders’ workshop and from the Ministry of Fisheries data. Usually, at least two of the above tests are carried out, and if all yield negative results, fish would be considered safe to consume. However, supposing they are still in doubt about the test results, the older family members or parents usually take a risk and eat the fish. When there is an absence of toxic symptoms, only then are children or lactating and pregnant mothers permitted to eat them. In the case of positive signs of the presence of the perceived toxin, the fish is discarded immediately. 

### 2.5. Locating and Removing Toxic Fish Organs or Tissue

Fishing during the perceived ‘toxic season’ demonstrates a high risk-taking tendency of fishers. Removing potentially toxic organs from fish is commonly practised for moray eels and pufferfish. From our interviews, most fishers are knowledgeable in distinguishing the two species of moray eels: *G. javanicus* and *G. flavimarginatus*. They claim *G. javanicus* is potentially less toxic than *G. flavimarginatus*, and that the best season to harvest *G. javanicus* for human consumption is soon after the *balolo* season, i.e., January to May.

Despite the knowledge that some fishes could be perceived as toxic, especially if caught from the perceived toxic hotspots or during the perceived toxic season, they are still harvested by fishers, sold, and consumed in certain Fijian households and restaurants. This practice is particularly true for moray eels and pufferfish. The harvesting of these potentially toxic fishes is probably encouraged by fishers’ knowledge of and use of TEK. In other words, consumers may have confidence in the skill of fishers and chefs in locating and removing toxic fish organs from certain species of fish. According to fishers in Lautoka, a Western town in Fiji, the toxins in moray eels are located along the backbones, liver and viscera. Therefore, to remove the perceived toxins, the moray eel is first gutted, and the whole backbones are completely removed, followed by a thorough scrub and wash of the whole fish in hot water. In Lautoka, *G. javanicus*, with the toxins removed, are sold at the fish market and cooked and sold in restaurants. This information was highlighted at the stakeholders’ workshop and confirmed by one of the researchers who visited the Lautoka restaurants. She observed people consuming moray eels prepared as described above and cooked as part of dishes in some restaurants. 

### 2.6. Local & Traditional Treatments of FP

Local and traditional treatments for FP usually involve various herbal medicines (see Table 5). According to findings from the informal *talanoa* session in this study, these medicines are usually prescribed by traditional healers or individuals who have had fish poisoning. The stakeholders’ workshop and interviews revealed more than 11 traditional herbal medicines. An additional 8 plants are listed in the Ministry of Fisheries data alone. There were five (5) treatments (marked with an asterisk (*)) that were common treatments identified by both data sources, i.e., the Ministry of Fisheries and the stakeholders’ workshop. Coconut stood out as one of the most widely used plants for treatment. It was interesting to observe that different parts of the coconut (e.g., coconut water, the thick cream extracted from coconut flesh, and charcoal made from burnt coconut shells) have been prepared and used in various ways.

The traditional herbal medicines used are mainly water-based and are prepared from either the leaves, fruit, barks, or roots of available plants. The leaves are the most dominant parts of the plants used to treat FP and are mostly taken orally. According to the stakeholders, traditional herbal remedies are often used to treat CP symptoms, especially in areas where medical facilities are inaccessible. These local and traditional medicines have been used to either induce vomiting to remove toxins and the food consumed or to induce diarrhoea to quickly excrete toxins through stool (if toxins have been digested). Drinking a good amount of concentrated coconut milk is an example of a local remedy that is expected to either induce vomiting or induce diarrhoea in the hope of quickly excreting the food containing toxins. Based on local knowledge, the thick coconut cream binds the toxin and quickly excretes it through the digestive tract in the form of watery stools. Similarly, consuming powdered coconut shell charcoal mixed with coconut water has also been practised, as charcoal is a toxin-binding agent. The adsorptive properties of activated charcoal that allow their use in treating intoxications are well reported in the scientific literature [25,26,27].

It was highlighted that traditional remedies used in communities depended on the plants available in the local area and that different communities have herbal medicines prescribed to FP victims. It was also confirmed that due to the prevalence of FP in maritime communities, people share the most effective remedies for treating FP. This sharing of treatment TEK is helpful because most treatment options and the plants used have the same common and scientific names throughout the country, even though local names might differ. Stakeholders revealed that, based on the recovery rate from FP, some traditional medicines were more effective than others. However, it must be borne in mind that recovery from any form of intoxication is based on several factors, including the concentration and amount of toxins ingested and the victim’s age and general health status.

## 3. Discussion

This research identified and documented TEK that has been used in the management of FP in Fiji. These TEK have shaped the perceptions of fishers and fish consumers alike on the various forms of seafood toxicity and their environment, how to monitor risks associated with suspected toxic fish species, and treatment options for intoxicated victims. The six locals and TEK identified were further categorised into two lines of defence approaches: preventative and treatment strategies. The preventive strategies were further grouped into pre- and post-harvest (see Figure 2). The preventive strategies included (i) the identification of perceived *iqoliqoli* hotspots with potential toxic fish, (ii) knowledge of the season when potential toxic fish are more predominant, (iii) knowledge of and ability to identify potential toxic fish species, (iv) performing folk tests for detecting potential toxins in fish, and (v) locating and removing perceived toxic fish organs. These preventive TEK can be roughly grouped into pre-harvest and post-harvest strategies. The treatment strategy included the use of herbal medicines.

One interesting observation from the identified TEK is the heavy reliance on prevention rather than treatment. This is evidenced by the fact that there are five (5) preventive TEK strategies and only one treatment strategy. This finding corroborated the growing awareness of how, in ecological preservation, indigenous knowledge systems focus heavily on prevention [28].

### 3.1. Preventive TEK Systems

#### Identification of Fishing Hotspots with Potentially Toxic Fish

Fishers in Fiji consider the geographic and environmental factors of the *iqoliqoli* before they go out fishing. In particular, they consider wind direction, as this is believed to influence the quantity of the potential harvest. Knowing about the potential ciguatera hotspots is another TEK that bar and ban fishers from fishing unsafe ciguatoxin-localised reefs. The use of local and TEK in identifying potentially ciguatera-prone hotspots has a precedent in the published literature, as it has been used by Raab et al. [7] in a study in Puerto Rico.

There are many ciguatera hotspots in Fiji, but the two common ciguatoxic-localized reefs are *Senimuna* reef in Kadavu and *Kabara* reef in Lau. The locations of Kadavu and the Lau group of Islands are shown in Figure 3.

All fish harvested from these two reefs are considered potentially toxic all year round. This information is widely known in the local communities near those reefs. The study participants in this research stated that, in Kadavu, there is no fishing in the known ciguatera hotspots at *Senimuna* reef because of this. Moreover, during a site visit to the *Senimuna* reef by some of the authors of this study, an abundance and diversity of colourful soft corals (*bulewa*) and sponges with patches of bleached hard corals were found (see Figure 4). The presence of *bulewa* may indicate a high risk that fish there are perceived toxic. A scientific inquiry is required to confirm the diversity of marine toxins in the reef.

Coral bleaching has also been associated with a high incidence of CP. This is because an increase in seawater temperature causes coral bleaching, which later kills the corals. Dead corals provide a large surface area for the growth of algae [23], becoming the ideal hosts for ciguatoxigenic organisms [29]. Coral bleaching is widespread in various *iqoliqoli*, including the *Kabara* reefs in Lau [30]. In their survey of the *Kabara* reefs, Areki and Fiu [30] observed a high prevalence of coral bleaching and coral mortality. They also observed that the coral growth and recovery were slow. No wonder our stakeholders indicated *Kabara* as a potential ciguatoxin–localised reef where all fish are toxic. Similarly, the *Senimuna* reefs in Kadavu have also been affected by coral bleaching, as observed by Cumming et al. [31]. Coral bleaching may have contributed to the high toxicity of the reef environment due to the growth and abundance of toxic dinoflagellate in the area.

The study participants pointed out reefs with inland catchments in Udu Point (Vanua Levu) that had proximity to the copper mining areas, and the mahogany plantations on Cicia in the Lau Group of Islands yielded most of the FP cases in the area. They cited a case in the 1990s where two people died from eating turtle meat from a mining village in the area. The interviewees believed that FP occurs when the relationship between the environment, land, and people is disrupted. This is a strong sentiment shared by most Fijians. The interviewers believed that heavy rain drained toxic wastes from the mine, which, together with dried mahogany leaves from the inland catchments, made their way into the sea and contaminated the seawater, causing coral mortality, which later provided the habitat for algal growth. Additionally, the possibility of copper poisoning cannot be ruled out.

Some fishers also believed that increased levels of perceived ciguatoxin-containing fish in certain potential hotspots could be due to disturbances caused by shipwrecks and wild weather patterns. For example, during the in-depth interview at Vanua Levu, it was brought to bear that, in 1990, some unicornfish (*Naso hexacanthus*) and surgeonfish (*Ctenochaetus striatus*) caught in the Udu Point waters were perceived as toxic following a hurricane. These local and TEK shared by participants agree with studies done elsewhere [32,33]. Studies by the *South Pacific Commission Expert Committee on Ciguatera* [33] also observed increased outbreaks of CP following the environmental changes in salinity, light intensity, and dissolved nutrients that control the micro-regionality of the *Gambierdiscuss* genus [33].

Interviewees also stated variations in the perceived toxicity of some fish species based on the location from where they are caught. For instance, the study participants mentioned how the longface emperors (*L. microdon*) that were never toxic in the past have become toxic on the Cicia reefs in the Lau Group of Islands. These puzzling facts are something that the TEK could not explain. Nevertheless, variations in climate and disturbances on reefs and corals may have contributed to the wide distribution and growth of marine toxin producers such as dinoflagellate [5,34,35,36], and some of these vectors might be consumed by various fish, leading to toxicity in the latter.

### 3.2. Knowledge of the Perceived Season When Toxic Fish Are More Predominant

Knowing the perceived seasonality of reef fish toxicity is another important local TEK that fishers use to manage FP. Information gathered from the stakeholders’ workshop and confirmed by the interviews revealed that FP incidences occur mainly during the warmer months (i.e., November–April). These warmer months coincide with two major ecological events: the hurricane seasons and the mass spawning of edible sea worms (*balolo*) [37]. Moreover, the luxuriant growth of macroalgae such as *Sargassum* spp., seagrass, and toxic microalgae occurs in Fiji during the warmer months [32], when seawater temperatures and nutrient levels are relatively high.

Findings gathered from the study participants show that the fish species generally implicated in FP during the *balolo* spawning season and warmer months are the emperors, snappers and groupers, while barracuda and moray eels are known to be toxic throughout the year. However, Bagnis [38] observed no direct causal link between the increased incidence of FP reported during the *balolo* season (October to November) in 1975 and theorised that ciguatoxicity may have been due to the trophic alterations of the environment following the release of a quantifiable amount of organic matter, together with other natural, mechanical, physical, chemical, and biological aggressions, such as reef disturbance from extreme events. This discrepancy in the findings of Bagnis [38] and those of this study can be explained when one considers that Bagnis [38] studied CP, while the responses from the current study centred on FP in general, without stratification into CP and other marine toxins. Moreover, there have likely been drastic changes in weather patterns between pre-1976 and now. These climatic changes may have caused ecological changes in the presence of FP (including CP) and the *balolo* season. *Gingold* et al. [39] also observed the associations between climate variability and CP incidence and suggested that when all other variables are equal, climate change tends to increase the burden of CP.

The seasonality of CP aligns with the warming of seawater temperature, especially in the warmer months when both *balolo* and corals are spawning and the growth of macro and microalgae is occurring. Hales et al. [40] supported this correlation, while Xu et al. [36] suggested a strong correlation between the rise in the incidence of CP and the increase in sea surface temperature during El Nino. A study in Tahiti also supported the association between the seasonality of CP and the maximum abundance of *Gambierdiscus* spp. population at the beginning and the end of the hot season [29], while in the Caribbean, a high incidence of CFP was associated with a warm sea surface temperature in September [41]. Further studies related to changes in seawater conditions have been correlated with the increase in seawater temperature, which has been associated with the increased distribution and growth of toxic algae and *Gambierdiscus* spp. [41,42]. Tester et al. [43] argued that a sea surface temperature of ≥29 °C during the hottest month in the Caribbean appeared to correlate with a high incidence of FP. This value appears to be similar to the hottest sea surface temperatures in Fiji, which vary between 29.6 and 30.5 °C [31]. Warmer conditions provide the ideal temperature for the rapid growth of dinoflagellates, increasing their likelihood of being grazed on by herbivorous fish and bioaccumulating up the food chain.

### 3.3. Knowledge and Ability to Identify Toxic Fish Species

A total of 34 fish were identified as potentially toxic (see Table 1) by the study participants in the current study. In this list, the three most implicated fishes (based on MoF survey data) were moray eels, snapper, and pufferfish. This information is consistent with reports elsewhere that snappers (*Lutjanus* spp.) and moray eels (*Gymnothorax* spp.) are the most widely implicated species for FP in the Pacific Islands [23,44,45,46]. The list of potentially toxic fish species and fish families identified in this study appear to be the same toxic species found in Australia [47,48], French Polynesia [44], New Caledonia [44], Cook Islands [45], Kiribati [49], New Zealand [50], and the other Pacific Island territories. Rongo and van Woesik [45] identified 48 toxic fish species in Rarotonga, Cook Islands, while more than 300 fish species have been reported to be linked to CP cases in all the Pacific Islands. Additionally, several ciguatoxin-containing fish in Table 1 have a carnivorous feeding habit. Examples of these fishes include barracuda, parrotfish, moray eels, Spanish mackerels, snappers, groupers, amberjack, mackerels, triggerfish, hogfish, sturgeon fish, kingfish, coral trout, and sea bass [23,43,49,51]. Additionally, the type of marine toxins in these fishes will differ. For example, *H. quadrimaculatus* is a well-known palytoxin fish [52], while *Arothron stellatus* is a tetrodotoxin-containing fish [53]. In contrast, most of the other fishes are ciguatoxin-containing fish.

An interesting finding from the current study was that study participants reported that some potential toxic fishes from potential ‘toxic’ *iqoliqoli* hotspots were found to be non-toxic in other fishing areas. For instance, they cited the fact that coral trout (*Plectropomus* spp.) caught on Cicia reefs did not show any toxicity symptoms. Yet, the same fish species in Onoi-Lau and Wailevu in Vanua Levu were toxic. Similarly, the study participants mention how mangrove red snapper (*L. argentimaculatus*) is perceived to be toxic in Macuata but not on the Cicia reefs in the Lau group.

It was also revealed that certain species of fish that were not toxic in the past have become toxic today. According to the study participants, the longface emperor (*L. microdon*) has never been known to be toxic on the Cicia reefs but appears to become toxic in recent times. It must be borne in mind that the toxicity of individual fish depends on several factors, such as time spent bioaccumulating toxins and the number of bioavailable toxins in the source organism [54]. Additionally, not all fish (even those known to be toxic) will accumulate toxins to levels that are toxic to humans [55]. Therefore, it is possible for fish species that were non-toxic in the past, or those found in non-toxic *iqoliqoli* hotspots, to be toxic in the present time, irrespective of location. This information shows a limit to how TEK can identify potentially toxic fish species, especially when identification is based on morphological observations rather than genetic analysis.

### 3.4. Performing Folk Test for Detecting Toxins in Fish

The folk tests used in rural communities in Fiji for detecting potential toxins in fish are shown in Table 4. Indeed, marine toxins such as ciguatoxin, palytoxin, and tetrodotoxin cannot be detected by physical inspections of the fish but only through scientific chemical analysis. However, the study by Darius et al. [56] studied two folk tests, namely, the rigour mortis test (RMT) and bleeding test (BT), and scientifically validated them against the receptor binding assay for detecting ciguatoxic fish. The authors reported a 55% and 69% efficiency for RMT and BT, concluding that the two folk tests can be used in parallel. However, these two tests are not practised in Fiji, hence their absence in Table 4.

Most folk methods may be discredited due to their lack of specificity and the regular occurrence of false-negative and false-positive results [56]. Some of the tests mentioned in Table 4 fall into this category. For example, there is currently no scientific basis for assuming a (potential) reaction between silver coins and marine toxins to generate discolouration on the coin or fish. Even if this reaction were possible, further studies would be needed to generate the conditions and parameters needed for accuracy, reproducibility, and precision. The other “bioassay” methods (e.g., exposure of fish to flies and feeding them to domestic animals such as dogs, cats and chickens) are rooted in common sense and reported in the literature [57,58]. However, using animals and even human adults in toxicity tests raises ethical questions. Moreover, these folk tests need to be scientifically validated to confirm their efficacy and use as a preventative measure for controlling FP.

### 3.5. Locating and Removing Toxic Fish Organs

Findings from the current study revealed that fishers who harvest the two highly perceived toxic species (moray eels and pufferfish) do so because they claim to have to identify the potential toxic organs in the fish and remove them, thus making the fish safe to eat. According to the study participants, this skill is developed over several years and usually consists of removing the fish’s head or gut, followed by thorough scrubbing and washing. The toxins in the fish gut, especially in the liver, could be 10–50 times higher than in the muscle tissue or flesh [49,57]. It was noted that some communities had experienced high toxicity levels from consuming moray eels. This came up during the Lau group workshop, where it was reported that certain family members were admitted to a health centre for treatment after they developed serious symptoms from eating moray eels. It is not clear whether the removal of the toxic organs and washing of the toxins were not carried out properly or whether some other factors were at play. It must be borne in mind that the flesh of the fish is normally consumed in large amounts. Thus, if the flesh is intoxicated with toxins, symptoms will still be experienced, even when other fish parts, such as the liver and heads, are removed.

For pufferfish, there may be variations in the concentrations of tetrodotoxin based on the species, types of tissue, geographic origin, sex and fish maturity [59]. In Japan, only licensed persons are permitted to handle and remove the toxic tissues from the list of approved pufferfish species before being served in restaurants for human consumption [59,60,61]. The study participants in this research asserted that most indigenous Fijian fishers are skilled in removing toxins from pufferfish. However, recent studies have shown that toxins in fish can be distributed in several parts of the fish tissue, including edible portions. For example, the flesh of pufferfish can be toxic [62], and in grouper (*Variola louti*), the flesh and head can contain the same levels of ciguatoxins [63]. Insights from these studies mentioned earlier throw into doubt the validity of removing toxic organs as a management strategy for FP. Further scientific studies are required on the effectiveness of this TEK in removing toxins from fish.

### 3.6. Use of Herbal Medicines to Treat or Manage Fish Poisoning Symptoms

The only TEK treatment strategy mentioned by the study participants was using herbal medicines to treat or manage fish poisoning symptoms. Typically, when fishers and consumers finally consume the toxic fish and experience symptoms of FP, they resort to using local and traditional treatment approaches as a first resort before visiting the health centres. Herbal medicines play an essential role in managing FP in the remote and rural areas in Fiji, from where access to health centres might be difficult.

It is also likely that FP cases in these remote areas are usually not reported. Long distances, poor road conditions, limited resources at the health centres, and the prevalent use of traditional herbal medicine in remote communities may be some reasons why some cases may go unreported. A list of herbal preparations used to manage FP in Fiji is shown in Table 5. In other Pacific Islands, numerous traditional herbal remedies are preferentially used to treat CP and FP [4,6,64]. In New Caledonia and Vanuatu [4], about 60 and 90 [65] plant species are used as traditional remedies in treating CP. Similarly, 24 plant species were used to treat ciguatera in French Polynesia. Among these plants, *Cocos nucifera*, *Punica granatum*, *Barringtonia asiatica*, and *Heliotropium foertherianum* were the most commonly reported plants for managing CP [64,66]. *Cocos nucifera* and *Barringtonia asiatica* are also commonly reported species for ciguatera in Fiji (see Table 5).

Other plant species in other countries had been tested for their anti-diarrheal, antispasmodic, anti-pruritic, or cardiac-tonic properties that had relieved some ciguatoxin symptoms in patients [23]. Some plant preparations have demonstrated the ability to reverse sodium channel activators’ effects in patients with ciguatoxin [65]. However, the herbal plants identified in Table 5 of the current study have yet to be scientifically tested to confirm their effectiveness and mechanisms of action in treating CP.

## 4. Conclusions

In this study, six local and TEK used in the management of FP in Fiji were identified, documented, and categorised into preventative and treatment strategies (see Figure 2). Information gathered in this research provides the baseline for further studies based on scientific validation of some of the local and TEK for FP management in Fiji. Moreover, this study opens up new opportunities for research in the chemical analysis of the presence and distribution of potential toxins in the identified fish and identifying the effectiveness of strategies such as removing toxic organs from fish and using herbal medicines to treat fish poisoning. Further research is also needed to understand the ecological factors that trigger the presence of ciguatera in hotspot areas or during certain seasons or the correlation between the *bulewa* and ciguatoxins.

Some form of knowledge sharing on the various TEK used in the management of FP among the various Pacific Island Small states is recommended. For example, although two folk tests (rigor mortis test and bleeding tests) have been scientifically validated and widely used in French Polynesia, these tests are not practised in Fiji. In connection with this, institutions such as the Pacific Community could coordinate the sharing of such information among the Pacific Island Small States, especially for TEK that has been scientifically validated.

## 5. Methodology

The data were gathered through various approaches, including a 2-day participatory stakeholder and key informants’ workshop, face-to-face interviews, in-depth interviews, and field observations. These activities were also part of a project entitled “Investigation of the incidence of fish poisoning in communities in Fiji”. The Ministry of Fisheries Ciguatera Survey raw data from 2010 to 2015 was used to validate and complement the data gathered in the current research. Ethics approval was obtained from the Fiji National Health Research and Ethics Review Committee of the Ministry of Health and Medical Services, with the reference number 2018.35.MP. The informed consent of participants was obtained, and participation was voluntary.

### 5.1. Participatory Stakeholder and Key Informants Workshop

A total of 22 participants that represented fishermen, holders of traditional knowledge, and ciguatoxin-poisoned victims from the four major divisions in Fiji (Central, Northern, Eastern and Western) participated in a 2-day workshop. The Ministry of Fisheries identified these participants through their fishery community network. The stakeholders of the Northern division were represented by Macuata province, while Muaivuso and Waiqanake districts and the Rewa province represented the Central division. The Western division was represented by Lautoka stakeholders, while the Eastern division was represented by its three district stakeholders of Lau, Kadavu and Lomaiviti. The Food Unit of the Ministry of Health and the Ministry of Itaukei Affairs also participated in the workshop. The participatory approach encouraged participants to freely discuss and express their experiences of local and TEK of FP, which were captured through group discussions, consensus, and presentations. Six separate sessions based on stakeholders’ divisions consisted of fishermen, holders of traditional knowledge, and ciguatoxin-poisoned victims for each group. A leader, a rapporteur, and a presenter for each group were selected to facilitate the discussion and to record and present their responses on the following three topics: (i) experience and knowledge of perceived poisonous fish (species) and major toxic hotspots or *iqoliqoli* sites and seasons; (ii) detection of toxic fish using folk tests and locating and removing toxic fish organs; (iii) the use of herbal medicines to treat and manage fish poisoning. Questions related to the three topics listed above were presented to each group after a PowerPoint presentation that introduced and explained CP.

### 5.2. Group Informal Talanoa

*Talanoa* research methods have been developed and are available and widely used by Pacific Islanders. It is similar to a narrative interview, an open, informal conversation between people sharing stories, thoughts and feelings. In essence, it is an ethnographic approach based on an open-style process of deliberation that is distinctive to Pacific people [67].

The face-to-face group *talanoa* sessions were conducted in three coastal villages in the Cakaudrove district of the Northern division, two coastal villages on Cicia Island in Lau of the Eastern division and two villages in Kadavu of the Eastern division. Three groups of about 6–8 people per *talanoa* group were conducted using the *talanoa* research methodology. All the *talanoa* sessions were informal and were undertaken in a working group context, such as during the mat weaving sessions for females and the food preparations for both females and males. Some information was also captured during the leisure hours after church service on a Sunday afternoon. The importance of creating the *talanoa* sessions to be as informal as possible was to allow more participants to contribute freely and allow stories to unfold naturally in their dialect, which was later translated and transcribed into English. Each *talanoa* session usually began with a question from the chair that triggered responses and discussions from participants within the group to share experiences of fish poisoning and contribute freely and openly. The purpose of the *talanoa* sessions was to provide new information and validate or confirm the information gathered from the 2-day stakeholder workshop mentioned above.

### 5.3. In-Depth Interviews

Seven separate in-depth interview sessions were carried out following the group informal *talanoa* sessions in three villages of the Cakaudrove district, two villages on Cicia, and two villages on Kadavu island. The interviewees were identified through the group *talanoa* sessions and interviewed in their own homes. A semi-structured interview questionnaire was used as a guide. Each interview lasted about 30 min and sometimes longer (i.e., 45 min). Other members of the interviewee’s family were present during the interview. Sometimes they would contribute to the conversation or be used by the researcher to validate certain data on the spot as a triangulation strategy. The interviews were recorded, and field notes were used to record data. Similar to the group *talanoa* sessions, the purpose of the in-depth interview was to validate, confirm or provide new information on the three major topics discussed at the stakeholder workshop.

### 5.4. Field Observations

Field observations were also conducted after the in-depth interviews to confirm information gathered from the workshop and group interviews. This was important to provide evidence of various information gathered from the workshop and to better understand the various FP local and TEK used and the settings in which they are applied. Data gathered were mainly in the form of field notes and pictures observed with ecological or environmental significance or important activities related to FP, including selling and consumption of toxic fish species, *iqoliqoli* hotspots, and fishing grounds referred to by the interviewees and participants of the group *talanoa* sessions.

### 5.5. Analyses of Ministry of Fisheries Ciguatera Survey Data 2010–2015

The Ministry of Fisheries Ciguatera Survey raw data from 2010 to 2015 were analysed to validate and complement information gathered from the current research. The triangulation was done by cross-checking information from the *talanoa* session, in-depth interviews, questionnaires, and field observations of local and TEK, such as fish species implicated in CP, their hotspots, seasonality, remedies and treatments.

The Ministry of Fisheries conducted the ciguatera survey as part of the Socio-economic and biological surveys of traditional fishing grounds (*iqoliqoli*) (see insert in Figure 1). These *iqoliqoli* were selected from 43 districts in Fiji that cover 1/3 (137 of 410) of the marine *iqoliqoli*. The survey used a semi-structured questionnaire that asked for information on the following: fish poisoning cases and experience, toxic fish species, associated *iqoliqoli*, seasonality, and its treatment.

### 5.6. Data Analysis

The workshop data were obtained from the consented group presentations based on the three topics: (i) experience and knowledge of perceived toxic fish species, major toxic hotspots, or *iqoliqoli* sites and seasons; (ii) detection of toxic fish using folk tests and locating and removing toxic fish organs; (iii) the use of herbal medicines to treat and manage fish poisoning. These were later summarised into common themes as shown in the result tables. The interview transcripts were manually coded, entered on MS Excel spreadsheets, and summarized into common themes as per workshop data. This helped in the confirmation of workshop data while also identifying new information. The participant observation data were matched with the interview data that identified evidence of the information provided in the interview. The Ministry of Fisheries data were provided to us in a coded excel spreadsheet that helped us match the relevant themes to our research themes. At the end of this process, themes were drawn for interpretations and discussions.

## Figures and Tables

**Figure 1 toxins-15-00223-f001:**
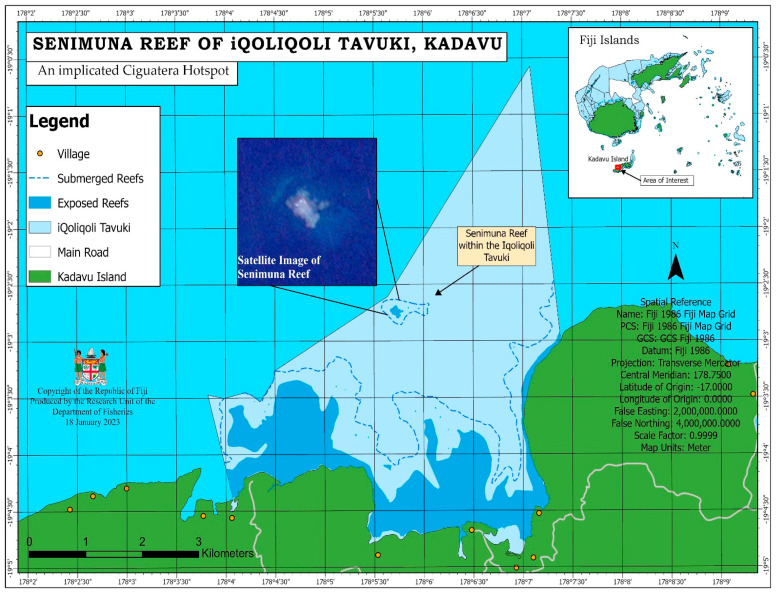
A map of *Senimuna* reef of *iqoliqoli* Tavuki, Kadavu, Fiji. Insert map of Fiji Islands (Source: Fiji Ministry of Fisheries).

**Figure 2 toxins-15-00223-f002:**
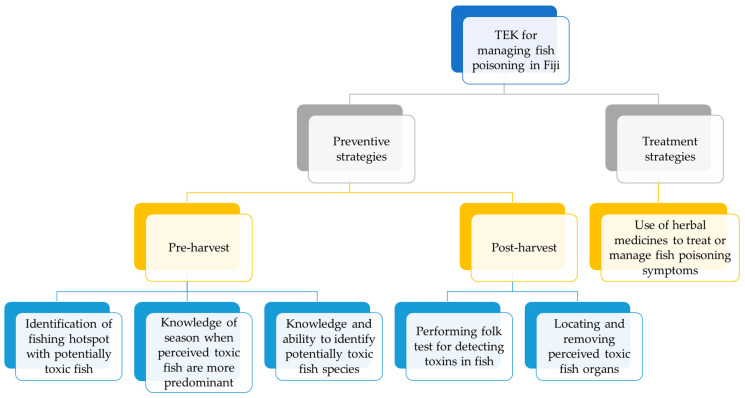
Schematic representation of the TEK identified in this study.

**Figure 3 toxins-15-00223-f003:**
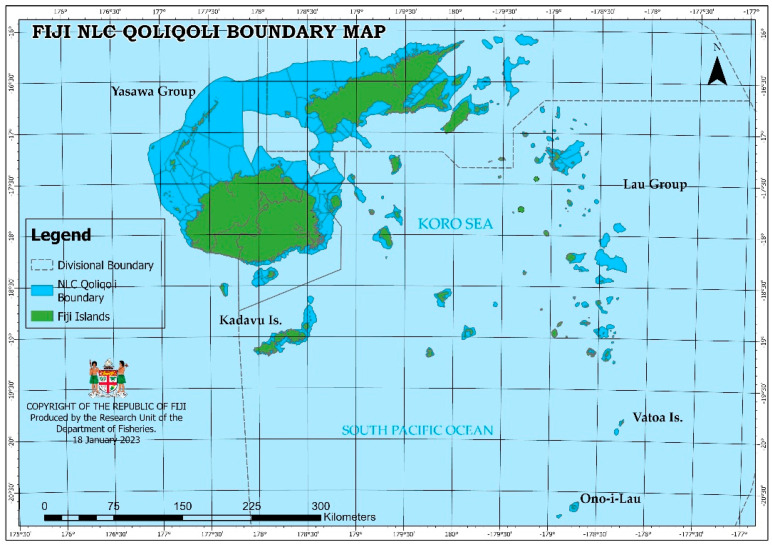
Locations of Kadavu and the Lau group of islands in the *iqoliqoli* boundary map of Fiji.

**Figure 4 toxins-15-00223-f004:**
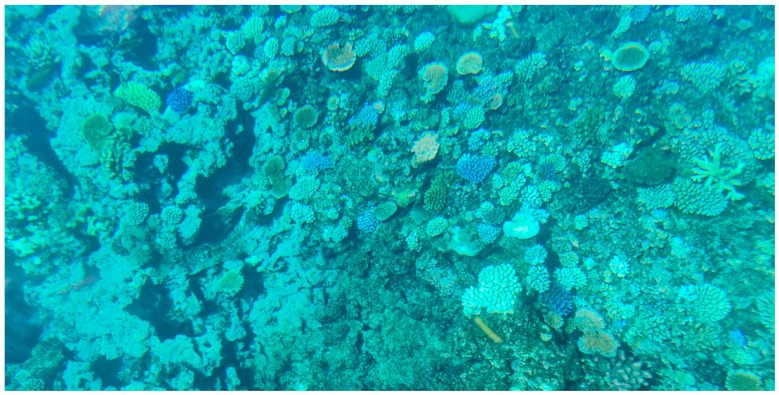
A photograph of colourful soft corals (*bulewa*) at the *Senimuna* Reef.

**Table 1 toxins-15-00223-t001:** Potential toxic fish species with implicated *iqoliqoli* (traditional fishing grounds) as identified by stakeholders and key informants at the workshop and interviews.

No.	Common Name (*Fijian Name*)	Scientific Name	Fish Family	Feeding Habits	Province	*Iqoliqoli*	Seasonality
1	Mangrove jack or mangrove red snapper (*Damu/Damu ni veitiri*)	*Lutjanus argentimaculatus*	Lutjanidae	Carnivores	Macuata		Unknown but sporadic
2	Moray eel (*Dabea*)	*Gymnothrorax javanicus*; and *Gymnothrorax flavimarginatus*	Muraenidae	Carnivores	Kadavu	* Senimuna reef	*G. javanicus* is perceived as toxic during *balolo* (edible seaworm) season; *G. flavimarginatus* is potentially toxic all year round. Toxin is located at the black spot of the backbone
					Lau		
					Macuata	Udu Point	
					Nadroga-Navosa	Nakavu	
3	Two-spot red snapper (*Bati*)	*Lutjanus bohar*	Lutjanidae	Carnivores	Kadavu	* Senimuna reef	Fish caught where there is *bulewa* (soft coral) are perceived to be toxic
					Lau	Lakeba	
					Cakaudrove	Udu Point	
					Lau	Bukatatanoa	
					Lau	Vanua Masi	
4	Stellate puffer fish (sumusumu)	*Arothron stellatus*	Tetraodontidae	Carnivores			
5	Leopard coral grouper (*Donu damu*)	*Plectropomus leopardus*	Serranidae	Carnivores	Kadavu	* Senimuna reef	Potentially toxic all year round
					Lomaiviti		
					Nadroga-Navosa	Yalewa Kalou	
6	Pink handle great barracuda (*Ogo buidromo*)	*Sphyraena sphyrona*	Sphyraenidae	Carnivores	Kadavu	* Senimuna reef	Potentially toxic all year round.
					Ra		Perceived toxic when the flesh becomes “stiff due to rigor”, or “*se*” in Fijian
					Lau	Kabara reef	
					Lomaiviti		
					Macuata		
7	Camouflage grouper (*Kawakawa*)	*Epinephelus polyphekadion*	Serranidae	Carnivores	Macuata		Unknown but sporadic
8	Blubberlip snapper (*Regurawa/Mesa*)	*Lutjanus rivulatus*	Lutjanidae	Carnivores	Kadavu	* Senimuna reef	Potentially toxic all year round;
					Lomaiviti		fish caught where there is *bulewa* are perceived toxic
					Macuata	Udu Point	
9	Russel’s snapper (*Kake/Sarau*)	*Lutjanus russelli*	Lutjanidae	Carnivores	Kadavu	* Senimuna reef	Potentially toxic all year round
					Lomaiviti		
10	White-spotted grouper (*Delabulewa*)	*Epinephelus caevuleopunctatus*	Serranidae	Carnivores	Kadavu	* Senimuna reef	Gonads are also perceived to be toxic.
					Lomaiviti		
					Macuata		
11	Giant trevally (*Saqa*)	*Caranx ignobilis*	Carangidae	Carnivores			Unknown and rarely occur
12	Striated surgeon fish (*Metu*)	*Ctenochaetus striatus*	Acanthuridae	Herbivores			Unknown but sporadic
13	Smalltooth emperor (*Dokonivudi*)	*Lethrinus microdon*	Lethrinidae	Carnivores	Macuata		During *balolo* season;
							mainly big-sized fish;
					Lomaiviti	Makogai	fish caught where there is *bulewa* are perceived toxic.
14	Longface emperor (*Leu*)	*Lethrinus olivaceus*	Lethrinidae	Carnivores	Macuata		During *balolo* season;
					Lomaiviti	Makogai	mainly big-sized fish;
							fish caught where there is *bulewa* are perceived to be toxic.
15	Titan triggerfish (*Cumu*)	*Balistoides viridescens*	Balistidae	Carnivores			Unknown and rarely occur
16	Bluetail mullet (*Kanace*)	*Valamugil engeli*	Mugilidae	Carnivores			Unknown but sporadic
17	Humphead wrasse (*Varivoce*)	*Cheilinus undulates*	Labridae	Herbivores			Unknown but sporadic
18	Reef whitetip (*Qio*)	*Triaenodon obesus*	Carcharhinidae	Carnivores			Unknown and rarely occur
19	Floral wrasse (*Dranikura*)	*Cheilinus chloroums*	Labridae	Herbivores			Unknown but sporadic
20	Porcupinefish (Sokisoki)	*Epinphelus lanceolatus*	Diondontidae	Carnivores			Unknown but sporadic
21	Pacific yellowtail emperor (*Sabutu*)	*Lethrinus atkinsoni*	Lethrinidae	Carnivores			Unknown but sporadic
22	Comet grouper (*Votoqaninubu*)	*Epinephelus morrhua*	Serranidae	Carnivores			Unknown but sporadic
23	Bluestripe herring (*Daniva*)	*Herklotsichthys quadrimaculatus*	Clupeidae	Omnivores	Macuata		Unknown but sporadic
					Lomaiviti	Gau	
24	Bluespine unicornfish (*Ta*)	*Naso unicornis*	Acanthuridae	Herbivores			Unknown and rarely occur
25	Thumbprint emperor (*Kabatia*)	*Lethrinus harak*	Lethrinidae	Carnivores			Unknown and rarely occur
26	Many spotted sweetlips (*Sevaseva*)	*Epinephelus polyphekadion*	Serranidae	Carnivores			Unknown and rarely occur
27	Crescent perch (*Qitawa*)	*Terapon jarbua*	Terapontidae	Carnivores			
28	Yellowstripe goatfish (*Ose*)	*Mulloides flavolineatus*	Mullidae	Carnivores			Unknown and rarely occur
29	Barred-cheek grouper (*Donu Damu*)	*Plectropomus maculates*	Serranidae	Carnivores			Unknown but sporadic
30	Sixbar wrasse (*Tekuru*)	*Thalassoma hardwicke*	Labridae	Herbivores			Unknown but sporadic
31	Spangled emperor (*Kawago*)	*Lethrinus nebulosus*	Lethrinidae	Carnivores			Unknown but sporadic
32	Topsail drummer (*Guruniwai*)	*Kyphosus cinerascens*	Kyphosidae	Herbivores			Unknown but sporadic
33	Whitespotted grouper (*Kawakawanitiri*)	*Epinephelus coeruleopunctatus*	Serranidae	Carnivores			Unknown but sporadic

* All fishes caught in the Senimuna reef in Kadavu are perceived to be toxic.

**Table 2 toxins-15-00223-t002:** The feeding habits and families of the potentially toxic fish identified during stakeholder workshops and interviews.

Feeding Habit, and Family	Number of Fish Species	Percentage
**Carnivores**	**26**	**79%**
Balistidae	1	3%
Carangidae	1	3%
Carcharhinidae	1	3%
Diondontidae	1	3%
Lethrinidae	5	15%
Lutjanidae	4	12%
Mugilidae	1	3%
Mullidae	1	3%
Muraenidae	1	3%
Serranidae	7	21%
Sphyraenidae	1	3%
Terapontidae	1	3%
Tetraodontidae	1	3%
**Herbivores**	**6**	**18%**
Acanthuridae	2	6%
Kyphosidae	1	3%
Labridae	3	9%
**Omnivores**	**1**	**3%**
Clupeidae	1	3%
**Grand Total**	**33**	**100%**

**Table 3 toxins-15-00223-t003:** Perceived toxic fish species implicated in 43 districts. Data from the 2010–2015 Ciguatera Survey Data of the Ministry of Fisheries, Fiji.

Common Names *(Fijian Name*)	No. of Districts Reporting This Fish in the Survey Data
Barred-cheek grouper (*Donu Damu*)	1
Blubberlip snapper (*Regurawa*)	8
Bluespine unicornfish (*Ta*)	1
Bluestripe herring (*Daniva*)	1
Bluetail mullet (*Kanace*)	3
Camouflage grouper (*Kawakawa*)	10
Comet grouper (*Votoqaninubu*)	1
Crescent perch (*Qitawa*)	1
Floral wrasse (*Dranikura*)	2
Giant trevally (*Saqa*)	6
Humphead wrasse (*Varivoce*)	3
Leopard coral grouper (*Donu damu*)	12
Longface emperor (*Leu*)	4
Mangrove jack or Mangrove red snapper (*Damu/Damu ni veitiri*)	32
Many-spotted sweetlips (*Sevaseva*)	1
Moray eel (*Dabea*)	27
Pacific yellowtail emperor (*Sabutu*)	2
Pink handle great barracuda (*Ogo buidromo*)	12
Porcupinefish (*Sokisoki*)	2
Reef whitetip (*Qio*)	3
Russel’s snaper (*Kake*)	8
Sixbar wrasse (*Tekuru*)	1
Smalltooth emperor (*Dokonivudi*)	6
Spangled emperor (*Kawago*)	1
Stellate puffer fish (*Sumusumu*)	13
Striated surgeon fish (*Metu*)	6
Thumbprint emperor (*Kabatia*)	1
Titan triggerfish (*Cumu*)	3
Topsail drummer (*Guruniwai*)	1
Two-spot red snapper (*Bati*)	21
White-spotted grouper (*Delabulewa*)	8
White-spotted grouper (*Kawakawanitiri*)	1
Yellow stripe goatfish (*Ose*)	1

**Table 4 toxins-15-00223-t004:** Local Folk Tests to Identify Potentially Toxic Fish.

Local Remedies	Changes Observed
Cooking suspected toxic fish with silver coins	The fish is perceived as toxic if the coins or any part of the fish discolour. If no discolouration is observed, the fish is safe.
Expose suspected toxic fish to flies	If flies avoid resting on fish, the fish is perceived as toxic, but if flies rest on fish, the fish is safe.
Animals such as cats or dogs are fed a piece of suspected toxic fish, especially the viscera or flesh, and then monitored for reactions	If the animals get sick or die, the fish is perceived as toxic, and when there is a reaction and the animals are not affected, the fish is safe.
Suspected toxic fish is cooked and consumed only by older family members or parents	If older people or parents get sick, the fish is perceived as toxic and should not be eaten by children.

**Table 5 toxins-15-00223-t005:** Major Local and Traditional Treatments of FP from Stakeholders, Confirmation Interviews, and Ministry of Fisheries Data.

Common/(*Fijian Name*)	Scientific Name	Part of Plant	Preparation
Stakeholders’ Workshop and Confirmation Interview Data			
* Screwpine (*Vadra*)	*Pandanus tectorlua*	Aerial roots	Extract is mixed with juice of mature coconut water.
* Pandanus (*Voivoilili*)	*Pandanus carlcosus*		
Mulberry (*Masi*)	*Broussonetia papyrifera*		
Unknown (*Kulukulu*)	*Dilenia biflora (A.Gray)*		
Coconut (*Niudamu*)	*Cocos nucifera*	Coconut juice	
* Coconut (*Niudamu*)	*Cocos nucifera*	Flesh	Extract with thick coconut cream
Fish poison tree or sea poison tree (*Vutuvutu* or *Vuturakaraka*)	*Barringtonia asisatica*	Nuts	Extract with water
		Bark	Extract with warm water
Wild Salvia or Red Salvia (*Rogodamu*)	*Salvia coccinea*	Leaves	Extract with water
Monarch fern (*Vativati*)	*Phymatosorus scolopendria*	Whole plant	Extract of whole plant with the juice of the husk of the green-yellow coconut.
Coconut (*Niudamu*)	*Cocos nucifera*	Green husk of yellow coconut	
* Pandanus (*Vadralili*)	*Pandanus tectorius*	Aerial roots	Extract with water
Sensitive plant (*Tubu ni mocemoce*)	*Mimosa pudica*	Leaves	Extract with water
* Coconut (*Niu*)	*Cocos nucifera*	Shell processed into charcoal	Charcoal pounded into powder and mixed with cold water
Ministry of Fisheries Data			
*Wiriwiri*	*Jatropha curcus*	Leaves	Extract with water
*Vulokaka damu*	*Vitex trifolia*		
Banana (*Jaina*)	*Musa* spp.		
*Vevedu*	*Scaevola saracea*		
*Mulomulo*	*Thespesia peoulnia*		
*Kura*	*Morinda citrifolia*		
*Drala*	*Erythrina variegate*		
*Dawa loa*	*Pometia pinata*	Bark	Extract with warm water

***** These are common treatments as identified from the Ministry of Fisheries data and stakeholders’ workshop.

## Data Availability

Not applicable.

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
