# Peer review of "Local and Traditional Ecological Knowledge of Fish Poisoning in Fiji"

_toxins, 2023, doi:10.3390/toxins15030223_

Round 1

Reviewer 1 Report

Line 10 Six TEK topics related to FP were .....

Lines 16-17 I am looking for parallel presentation of of the two lines of defense. At this point your reader does not know if you accept the Folk tests or will examine those and make recommendations. Please reword this sentence to make the reference to Folk treatments clearer. 

Line 52 those who dwell

Lines 56-58 Scombroid or histamine fish poisoning is caused by bacteria when fish, especially those with dark flesh, are poorly handled post-harvest and stored at temperatures above 4C (Laka, Solo and Ishigaki, 2015).

Line 60 Delete "and others". That adds no information. 

Lines 65-67 This sentence needs to be reworded. It is confusing.

LIne 69 I suggest updating the reference rather than using a 1989 one. There has been much more recent research on Gambierdiscus.

Line 79 G. toxicus is one of the least toxic of the Pacific Gambierdiscus. G. toxicus, while the type species for the genus, was not described from a single but multiple cells, not all of which were G. toxicus. An epitype was named for this species (Litaker et al. 2009 - Phycologia). I suggest G. polynesiensis be used instead of G. toxicus, as G. poly is widely acknowledged and documented as the major source of CTX in the Pacific. Any designation of Gambierdiscus species prior to 2009 should be viewed with caution.

Lines -83 I note that you've missed two major reviews that are relevant to your MS. Both of open-source papers, one by Hallegraeff et al 2021 and the second by Tester et al. 2020.

Line 101 distances

Lines 108 & 108 Is the term "mild cases" appropriate rather than CP patients. Do cases = a person?

Line 124 illnesses 

Line 139- 140 This sentence needs to be rewritten. ... consumed and can be explained by local and TEK. 

Line 149 Delete second "all".

Line 151 Use confirmational rather than confirmed interviews.

Line 149 Table 1. Define balolo and bulewa in the table description. I know these terms have been defined in the text, but consider adding them to the table title so the table can be a "stand alone" piece of information for those not reading the text.

Line 161 Table 2 Please explain what 1/3 iqoliqoli means. Are toxic fish found in one third of the habitats or ??? Please clarify.

Line 164 I suggest this information (74% of 43 districts (1/3 of iqoliqoli in Fiji) be copied into Table 2 description so if the paragraph starting on line 163 is not before the table in the final printing of this paper, the reader will not be confused. Again, this just speaks to the table and table description or title being a "stand alone" product of this MS.

Line 170 Delete when and replace with where.

Line 208 "their successful removal"...

Line 220 replace that with who.

Line 262 "and if all yield negative results, fish would be considered safe to consume."

Line 281 Replace that with who.

Line 288 "where CP victims, after eating moray eels, were admitted....

Line 328 "quickly excreting the food containing toxins."

Line 360 Do you mean validating rather than violating?

In the discussion about toxic fish (turtle) from a mining area, how do you know the toxicity was not from mine tailing (contamination)? 

The discussion is a trove of information that might not have been generated except by the methods outlined in this paper. It is truly rewarding to see the local population take FP in hand for review and suggestions on how better to manage the problem. The community outreach was likely far more effectively done by local staff than a group unfamiliar with the day-to-day lives and livelihoods of the local populace. I was delighted to read this paper and be able to learn things not found in more scientifically based papers.

I recommend a minor revision that includes a careful look at the English and reduces the redundance in the discussion section. Perhaps section headings could help better organize and streamline information in the discussion that is already in other section of the MS. Overall, though, it was a pleasure to read and understand the different types of information coming from the different and varied sources included. Congratulations!

Author Response

Please see the attachment, thanks.

Reviewer 2 Report

This article has poor scientific information content, since the main data presented in the MS are based on surveys. The MS contains many logical errors without any scientific data, and below I give only some of them. In general, an article is not suitable for publication in Toxins journal.

The MS points out that 33 species of fish caught off the coast of Fiji are toxic - but does not indicate how these species were identified: whether genetic markers were used, or only morphological ones.

There is no data on how exactly toxin was detected: does outhors or anyone else used scientific methods for toxin detections?  The toxin detection methods proposed by the authors (see Table3)  are not suitable.

The authors proposed to treat patients with fish poisoning by traditional medicine. It does not correct - Patients with poisoning have to be treated in hospitals under the supervision of doctors.

Line 5 "Fish poisoning (FP) threatens human health, trade and livelihood" This statement corrects only for some regions of tropic and subtropics regions.

Abstract does not reflect conducted research and must be completely rewrite. The same situation with “Keywords” and “Key contribution”

Line 41 Only some «Pacific region» are exposed of “Fish poisoning”.

The Introduction is a enumeration of unrelated facts about "Fish poisoning" around the world. I recommend to rewrite the introduction section to focuse on on the Fiji or/and related regions.

The authors identified six local and TEK. But the category "known toxic fish species" and the category "folk tests that detect toxic fish" are the same thing - they are a list of toxic species

In Line 215 The authors say about "410 traditional fishing grounds" and refer to "Figure 1" which shows "fish families".

Line 226: “detailed data (not shown)” – the authors provide unconfirmed data. Please add this data to the supplementary file.

Figure 2 and Figure 3 Does not carry any scientific information. Please delete these figures.

Table3 - The methods shown in Table3 are only generalized opinions of the islanders, and do not have any scientific confirmation.

Title of “2.5. Locating and Removing Venom Glands from Fish” is incorrectly named. The tissue and cellular localization of the toxin in toxic fish is only partially related to the skin glands. In the liver, the toxin is contained in the cytoplasm of hepatocytes, and in the gonads - in the cytoplasm and nuclei of the eggs. Check out the recent review on toxin localization in puffer fish (see https://doi.org/10.3390/toxins14080576) - this review will help to better describe this chapter.

The authors give detailed data on the use of traditional medicine in the treatment of poisoning. However, traditional medicine does not have a scientific basis. Nevertheless, in case of poisoning with toxic fish, patients should contact with medical specialists.

Author Response

Please see the attachment, thanks.

Reviewer 3 Report

Te paper entitled ‘Local and Traditional Ecological Knowledge of Fish Poisoning in Fiji’ complies very useful information about folk methods to prevent and treat fish poisoning. It has been performed through exhaustive research based in workshops and interviews with the knowledge holders as well as by contrasting data by field observation and scrutiny of official sources. The paper can be relevant for scientific fields in toxicology, particularly in the research of marine toxins such as ciguatoxins, palytoxins and tetrodotoxins. Therefore, I recommend the article for publication in Toxins and congratulate the authors for the great job.

Despite this, there are two main issues that should be revised and corrected:

1st, there is a misinterpretation about ‘venom’ and ‘venom gland’ which (I think) also led to a misunderstanding of the (arguably) effectiveness of viscera removal as a method to avoid fish poisoning (FP).

2nd, although the writing is good, with minor format errors, the paper, could be better structured. Mainly in the discussion section, some parts are very repetitive and sometimes paragraphs seem not to follow a logical order. I would suggest that the authors order the main ideas and topics and develop them coherently throughout the document.     

Detailed comments and remarks:

Throughout the document:

Remove the word ‘venom’, the correct term is toxin/toxic (venom is injected, usually produced by the same organism, toxin is eaten). Also, change ‘glands’ for ‘organs’ or ‘viscera’ or ‘animal tissue’ or another word. A gland is for secreting compounds. When you speak about ‘venom gland’ you are evoking a specific organ that secrets venom (poisonous compound that is given off by biting, stinging…in a more or less voluntary way). This is not the case you are dealing with in the article. Fish poisoning due to ciguatoxin, palytoxin or tetrodotoxins consumption are transferred to fish through the food web and accumulated mainly in fish viscera (fish liver for example, in the case of ciguatoxins). That is why these organs are usually more toxic than flesh, not because they have a specific role in producing or secreting venoms.   

Abstract

12. Spawning season of toxin vectors: I understood from the text that the spawning season of a worm was concomitant with the highest episodes of FP, but it does not mean that the worm can transfer the toxins to fish. I think it would be more accurate just mention that you identified the seasons when the risk of FP is highest.

14, 15. Remove subscripts, as you already have the meaning embedded in the text.

17-18. Applying these TEK could stem the tide of fish poisoning in Fiji:  Is a good closure phase, but sounds too optimistic.  Maybe ‘The TEK collated in this work can be of help for local authorities to better identify the sources of toxicity and applying TEK preventive measures could contribute to stem the tide of fish poisoning in Fiji’ or something similar.

Keywords: ‘hotspots’ is a word with a very wide meaning; I would choose other more specific for your study, for example ‘marine toxins’.

Introduction

55. You have mentioned here scombroid FP, but barely mention it, later. If you did not identify any FP due to histamine poisoning, may be it would be worthy to comment this fact in the discussion section. Otherwise delete it from this section.

60-63. is the specie referred in this article common in Fiji? Although this sentence must be true for most species of pufferfish, not all of them follow this rule. Later in discussion you mention Arothron stellatus, which can burden toxins even in the flesh. Search for a reference that better support what you are exposing here.

65-67. This sentence is confusing. Palytoxins are produced by zonathid corals and benthic dinoflagellates . Certain marine organisms, including fish, feed on these PLTX producers.

70. As there are several species Gambierdiscus species with diverse toxicity, better refer to Gambierdiscus species. If you want to highlight one of them, G. polynesiensis is among the most toxic ones, but not G. toxicus. https://doi.org/10.2216/07-15.1, 10.1016/j.toxicon.2009.06.013

74-102 and 123-126. These paragraphs are related to ciguatera (CFP). You speak about ciguatera then FP in general and then CFP again, making the text a little bit disorganize. Paragraphs 74-102 may fit better in the discussion section (in fact you say that in the discussion. In 123-126 you speak about treatment antidote for CFP, what about the other FP?

I suggest organizing better the introduction regarding this part about FP. In the introduction, provide a wide perspective of FP in general and then go to specific summarizing the most important toxins/fish-borne diseases. No more than 5 lines for each one. For example:

-Ciguatera: vectors, mode of action, symptoms, treatment, prevalence in Fiji…

- TTX: vectors, mode of action, symptoms, treatment, prevalence in Fiji…

-PLTX: vectors, mode of action, symptoms, treatment, prevalence in Fiji…

- Histamine?

Or whatever other formula you prefer, but be consistent.

Results

Table 1 and Table 2, could be merged. Remove subscripts in captions.

Figure 1. What the y axis represent? Put titles in the axes, and units if necessary , remove hashes in the legend, and why did you take into account the paper Bagnis (1976) and no other (recent) literature.

179-187. For discussion section

203-209. You explain that in the next section. Focus on what are you speaking about to avoid repetition. This section is all about seasons.

210-211. ‘Based on the interviews, most fishermen are knowledgeable in distinguishing the two species of moray eels: G. javanicus and G. flavimarginatus for their levels of toxins.’ There is no way to distinguish moray eels for the level of toxin without a laboratory assays. You can distinguish the species which tend to accumulate more toxins, which in this case is G. flavimarginatus. Also, this paragraph must be mentioned in the section about species.

284-290. For discussion.

292-293. Not allways, and not in the case of the species you mentioned, be cautious. This result is 'according to the information that fishers provided'. This state did not reflect a result of your investigation, since you did not analyze pufferfish. This statement should have a reference and better go to the discussion section.

318. Ciguatera (CP) symptoms.

Discussion

The discussion is quite repetitive with results.  I think it could be clearer if the authors add subtitles in the same way than the results section. For example:

3.1. Potential Toxic Fish Species

Potential ciguateric fish,…..

Pufferfish…

Etc…

There is no need to use exactly the same titles but following more or less the same structure throughout the text will make the text more readable.

350-354. This is a good point, it could be mentioned earlier in the introduction (and the abstract, if possible) to highlight the significance of this work.

372-376. This is quite contradictory with the previous sentence. I the cases have dramatically increased it seems that the removing of viscera may not be so effective. Ciguatoxins and TTX can be certainly accumulated in fish muscle and the removal of viscera may not be a solution at all.

383-384. I would not afirm that this traditional knowledge can detect the toxin, as it has not been validated by laboratory assays.

397-404. This is more in line with the paragraph 372-75, organize the text by subjects.

406-410. The same mentioned in previous comments, be cautious and check bibliography about this, there are not such a general rule for all species and toxins.

420-422. The fact that the researcher did not get poisoned just confirm that the moray eel flesh did not bear toxins or not in such concentration to cause symptoms but did not confirm that the removing of viscera is effective. Be very careful with this kind of sentences because can lead to confusion to readers. The only way to know if a fish contain ciguatoxins  in any of their tissues is with an assay/bioassay or instrumental analyses.

423-424. Again, depending on the toxicity level, the toxin can be also in the muscle, so removing the viscera would not make any difference.

429-430. The same, check the reference.

433-437. Here, are you speaking about ciguatera of FP in general? Start speaking general and then go to specific.

440. ‘Based in literature’ Which one, add references.

444. Again, check the reference. In the paper you cite it is reported TTX in muscle of Arothron stellatus.

453-454. You have spoken about ciguatoxin previously, organize the text.

473. NZ?

475-481. This can be earlier in the introduction.

482-487. This paragraph is repetitive and not really accurate, delete it, please.

492. ‘Infected’ is not the correct word.

519. Refer to Gambierdiscus spp. since the taxonomy for this genus have changed since then.

550. Explain what kind of restrictions. If legal regulations exist, provide the reference.

556. Is this bulewa coral pertaining to the genus Palythoa? It would be nice to have this information. If it is may be more related to clupleotixism and palytoxins. If you do not have this information you could state it, but regarding FP-clupoteoxism, not CFP.

474. I don’t understand what unvalidated information is this. Anyway, if it is no valid, is it worthy to mention it?

575-581. It seems perfectly possible to me.

594. Couldn’t it be related to copper poisoning?

595-596. ‘Interviewees believed that CP occur when the relationship between the environment, land and people is disrupted’. This is a result from your study that was not previously mentioned.

613-614. Something similar was previously mentioned, order the main ideas.

635-637. Better mentioned in the introduction.

637-655. Repetitive with results.

659. Should be explained previously.

Author Response

Please see the attachment, thanks.

Round 2

Reviewer 2 Report

This article is not suitable for the Aim of Tixins journal. Toxins journal aim is "scientists to publish their experimental and theoretical results in as much detail as possible". In "Author Response File" the Authors tell that - "This paper is about the Local and Traditional Ecological Knowledge of indigenous communities gathered through surveys without any scientific experiments or evidence". The MS must be rejected. 

Author Response

We are aware of the scope of this journal. In fact, before submitting our manuscript, we first emailed an abstract and title to Marie-Yasmine Dechraoui Bottein to inquire about the suitability of this manuscript for the special issue. She replied and encouraged us to submit the manuscript. We also assume that once the manuscript was submitted, the Editor and Journal managers had a review before passing it on to the Reviewers. Thirdly, the other Reviewers gave feedback on the manuscript that presumes its suitability for this journal. Reviewer 1  described how this manuscript is important and "was delighted to read this paper and be able to learn things not found in more scientifically based papers." Reviewer 3 also remarked that the manuscript "has been performed through exhaustive research based in workshops and interviews with the knowledge holders as well as by contrasting data by field observation and scrutiny of official sources. The paper can be relevant for scientific fields in toxicology, particularly in the research of marine toxins such as ciguatoxins, palytoxins and tetrodotoxins."

We understand that Reviewer 2 would like to see quantitative data on the toxins in the manuscript. But, as we have made clear, the manuscript investigates and documents the use of local and traditional ecological knowledge in the prevention, management, and treatment of FP in Fiji. Based on the Editor's comments, we have qualified the article by clarifying that toxicity is "perceived" - to indicate the subjective nature of the study. We have also raised several points in the manuscript where the validity of the TEK is doubtful and why scientific (chemical) analysis will be needed. Our manuscript, therefore, lays the foundation for future studies that could involve quantitative analysis. From these comments, we believe our manuscript will be of interest to the broad readership of this journal.